# Exercise and Sirtuins: A Way to Mitochondrial Health in Skeletal Muscle

**DOI:** 10.3390/ijms20112717

**Published:** 2019-06-03

**Authors:** Katya Vargas-Ortiz, Victoriano Pérez-Vázquez, Maciste H. Macías-Cervantes

**Affiliations:** Department of Medical Sciences, University of Guanajuato, Campus León, 37220 Guanajuato, Mexico; kavati75@hotmail.com (K.V.-O.); vicpe@yahoo.com (V.P.-V.)

**Keywords:** exercise, health, mitochondria, PGC-1α, SIRT1, SIRT3, skeletal muscle, therapeutic target

## Abstract

The sirtuins form a family of evolutionarily conserved nicotinamide adenine dinucleotide (NAD)-dependent deacetylases. Seven sirtuins (SIRT1–SIRT7) have been described in mammals, with specific intracellular localization and biological functions associated with mitochondrial energy homeostasis, antioxidant activity, proliferation and DNA repair. Physical exercise affects the expression of sirtuin in skeletal muscle, regulating changes in mitochondrial biogenesis, oxidative metabolism and the cellular antioxidant system. In this context, sirtuin 1 and sirtuin 3 have been the most studied. This review focuses on the effects of different types of exercise on these sirtuins, the molecular pathways involved and the biological effect that is caused mainly in healthy subjects. The reported findings suggest that an acute load of exercise activates SIRT1, which in turn activates biogenesis and mitochondrial oxidative capacity. Additionally, several sessions of exercise (training) activates SIRT1 and also SIRT3 that, together with the biogenesis and mitochondrial oxidative function, jointly activate ATP production and the mitochondrial antioxidant function.

## 1. Introduction

The seven sirtuins present in mammals (SIRT1–SIRT7) are a family of class III histone deacetylases [1,2] that regulate histone proteins in specific lysine residues, promoting post-translational modification resulting in chromatin silencing and transcriptional repression. In addition to histone proteins, other proteins are also deacetylated by sirtuins, modulating their activity [3,4]. It is now known that the enzymatic activity of sirtuins is broader—they not only act as deacetylases, but also as ADP ribosyl-transferases, demalonylase and desuccinylase, all of which require binding with NAD [5,6]. In addition, it has been postulated that the function of sirtuins depends on their intracellular location [2].

The wide spectrum of cellular activities of sirtuins in the organism places them as therapeutic targets to combat metabolic, neurodegenerative and proliferative diseases. In this context, in underdeveloped and developed countries, diseases related to obesity, such as insulin resistance, type 2 diabetes and dyslipidemia have become a public health problem. Among the strategies to prevent or combat such diseases is exercise. Exercise positively affects the activity and/or expression of sirtuins, resulting in better oxidative metabolism efficiency, increased biogenesis and mitochondrial function, as well as maintenance of the antioxidant system [2].

In the last two decades, evidence has accumulated about the effect of exercise on the expression and/or activity of sirtuins. Much of this knowledge has emerged from research in animal and cellular models, and to a lesser extent from studies in humans, in most cases, in healthy humans. In this review we summarize the evidence about the effects of different types of exercise on SIRT1 and SIRT3 in skeletal muscle, mostly in healthy individuals, and the role of such sirtuins as regulators of the benefits of exercise.

## 2. Skeletal Muscle, Sirtuins and Exercise

Skeletal muscle not only functions to generate force and movement, it is now known that it is a tissue that acts as an endocrine organ secreting cytokines and transcription factors into the bloodstream, thus regulating the function of other organs. Furthermore, skeletal muscle is a metabolically active tissue with an important role in the maintenance of the metabolic homeostasis of the organism. Skeletal muscle comprises approximately 40% of the total body mass, and it is the major site for insulin-stimulated glucose up-take as well as the main energy consumer of lipid catabolism [7,8]. Metabolic flexibility, defined as the ability to switch between glucose and lipid oxidation, is crucial for skeletal muscle to maintain physiological functions and metabolic homeostasis [9]. Advances in knowledge about the molecular mechanisms underlying the proper functioning of skeletal muscle constitute a therapeutic benefit. The roles of sirtuins in regulating glucose and lipid metabolism, insulin sensitivity as well as function, and mitochondrial biogenesis have been widely investigated in skeletal muscle.

Physical exercise causes cellular metabolic stress, which impacts the sirtuins. In this context, the most studied sirtuins have been SIRT1 and SIRT3; SIRT1 is located intracellularly in the nucleus and SIRT3 is located in the mitochondria. The study models that have been used range from cellular cultures and rodents to humans. In 2009, Palacios et al. reported that mouse skeletal muscle SIRT3 responds dynamically to six weeks of voluntary exercise to coordinate the downstream molecular response. They showed that exercise increases the SIRT3 protein content, as well as the associated phosphorylation of cAMP response element-binding (CREB), the up-regulation of peroxisome proliferator-activated receptor gamma coactivator-1α (PGC-1α) and the citrate synthase (CS) activity. They also demonstrated that SIRT3 is more expressed in type I muscle fiber. The researchers found that in Sirt3 knockout mice, the phosphorylation of AMP-activated protein kinase (AMPK) and CREB and the mRNA of PGC-1α was down regulated. Hence, they suggested that these key cellular molecules could be important components of the biological signals regulated by SIRT3 [10]. Based on their findings, Palacios et al. proposed a model in which SIRT3 dynamically responds to exercise to potentially impact muscular energy homeostasis via AMPK and PGC-1α and suggested that given the dynamic role of SIRT3, it may be a therapeutic target that impacts human health and disease [10]. Hokari et al. (2010) confirmed the increase in the SIRT3 content of skeletal muscle in rats after 4 weeks of voluntary exercise or treadmill training, moreover, they reported that the SIRT3 content was downregulated in immobilized soleus muscle [11]. On the other hand, Bayod et al. (2012) reported increases in the protein content of SIRT1 and PGC-1 in the muscle of rats after 36 weeks of treadmill training, in addition, the authors observed an improvement in antioxidant defenses [12].

Knowing that the metabolic stress induced by exercise causes changes in the NAD+ and subsequently in the sirtuins, and after an interesting review, White and Schenk (2012) suggested that the increased ATP demand during exercise leads to an increase in the NAD+ level and NAD+/NADH ratio, which provides increased substrate for SIRT1 and SIRT3. The increase in ATP production is facilitated by SIRT3, which deacetylates and activates enzymes of the tricarboxylic acid (TCA) cycle, β-oxidation, and electron transport chain. At the same time, SIRT3 acutely reduces mitochondrial protein synthesis, which maximizes the availability of reducing equivalents for ATP production. On the other hand, in response to exercise, SIRT1 contributes to mitochondrial biogenesis through PGC1α-dependent and -independent mechanisms [13].

In 2012, Gurd et al. reported that SIRT3 is proportional to the PGC-1α protein and the oxidative capacity of the mammalian muscle, which means that there is more SIRT3 in the heart, followed by type I muscle fiber, and finally, in type II muscle fiber. They also confirmed that in the muscle of mammals, the subcellular localization of SIRT3 is the mitochondria [14] and that in response to acute exercise, there is no translocation of SIRT3 from the nucleus to the mitochondria as has been demonstrated in some cell lines [15]. Gur et al. noted that an increase in SIRT3 protein is not required for an increase in mitochondrial or PGC-1α content, since PGC-1α is upregulated in the absence of changes in SIRT3 after 5-amino-1-b-d-ribofuranosyl-imidazole-4-carboxamide (AICAR) treatment, suggesting that the SIRT3 content is regulated in an AMPK–independent manner. The researchers did not find a relationship between mitochondrial SIRT3 and fatty acid oxidation; however, they suggested the possibility that the post-translational activation of SIRT3, instead of the protein content, plays an acute regulatory role in oxidative metabolism in skeletal muscle in vivo [14].

The connection between the regulation of the production of ATP and SIRT3 was studied by Vassilopoulos et al. (2014); the data obtained by these researchers suggests that acetyloma signaling contributes to the homeostasis of mitochondrial energy through SIRT3-mediated deacetylation of ATP synthase protein. Their results demonstrated that the F1 portion of the ATP synthase complex contains multiple SIRT3-dependent reversible acetyl-lysines that are altered under various metabolic stress conditions, including exercise, in both mice and human samples [16].

Previously it has been suggested that SIRT3 carries out the antioxidant function by deacetylation of manganese superoxide dismutase (MnSOD) [17]. Shi et al. in 2018, reported their results on the SIRT3-MnSOD pathway and cognitive function. Their findings confirmed that the antioxidant effect of the SIRT3-MnSOD pathway is critical for the survival of neurons. They demonstrated that a high fat diet (HDF) alters this pathway by modifying the acetylation of MnSOD in the hippocampus, increasing the levels of oxidative stress and damaging cognitive function in mice. However, aerobic interval training attenuated neuronal apoptosis and improved cognitive function in mice with HFD through the positive regulation of the SIRT3-MnSOD pathway, and the reduction of oxidative stress levels. These findings would place SIRT3 as a new factor in the neuroprotective field [18].

In the same way, understanding the role of sirtuins as regulators of the benefits of exercise in human health has been pivotal to many investigations and obtaining greater knowledge. The section below is divided by subheadings. It provides a concise and precise description of the experimental results and their interpretation as well as the experimental conclusions that can be drawn.

## 3. Effects of Different Type of Exercise on the Sirtuins in Human Skeletal Muscle

Several investigations have focused on analyzing the effect of exercise on the increase of sirtuin expression, and measuring mRNA expression and/or the increase of protein or even the activity of sirtuin Although it is difficult to compare these studies, here are some of the important findings. Lanza et al. (2008) were pioneers in trying to explain the effect of exercise and influence of age on the expression of sirtuins. They undertook a cross-sectional investigation in which they analyzed healthy young and older subjects that were divided into sedentary and trained (subjects had performed at least 1 h of exercise 6 days per week over the past 4 years or longer). The research reported that regular endurance exercise partly normalizes age-related mitochondrial dysfunction, and the researchers observed that regular training, regardless of age, increased the activity of the CS and la expression of proteins involved in mitochondrial biogenesis including PGC-1α, nuclear respiratory factor 1 (NRF1), mitochondrial transcription factor A (TFAM) as well as SIRT3 [19]. Villanova et al. (2013) confirmed these results—they found that the deacetylase activity of sirtuins decreases after 40 years, but in athletes the activity of sirtuins is over regulated [20].

Another cross-sectional study was carried out by Koltai et al. (2018) with master athletes and sedentary subjects, all of whom were over 60 years old. Greater expression of forkhead box O1 (FOXO1) and SIRT1 mRNA was observed, as well as higher content of MnSOD and SIRT3 proteins in the master athletes compared with sedentary participants [21]. Only miR-7 (associated with inflammation [22]), was more expressed in sedentary subjects than in athletes. The authors concluded that performing lifelong exercise can lead to the suppression of inflammation related to sarcopenia and better fat metabolism. The higher content of SIRT3 is also compatible with the ATP production and the antioxidant capacity through MnSOD in the skeletal muscle of the master athletes compared to the control subjects [21].

The above is explained because SIRT3 deacetylates the medium-chain acyl-CoA dehydrogenase and acyl-CoA dehydrogenase, thus controlling the metabolism of fatty acids [23]. As mentioned above, SIRT3 interacts and deacetylates the F complex of ATP synthase directly increasing the production of ATP [16], In addition, deacetylation of FOXO1 by SIRT3 raises the expression of its target genes, such as MnSOD [24]. The latter is also deacetylated by SIRT3, promoting its antioxidant activity and decreasing reactive oxygen species (ROS) in mitochondria [25].

The investigations into the effects of physical exercise can be grouped by type of exercise: acute exercise (a single exercise load), exercise with successive loads, and exercise training (several sessions of high-intensity interval training (HIIT), aerobic or resistance exercise).

In analyzing Table 1, the first deduction could be that the single exercise load-induced changes in SIRT1 in skeletal muscle of healthy subjects are contradictory because in one investigation the protein content increased [26] and in another, mRNA expression remained unchanged [29]. However, increases in mRNA do not always lead to increases in protein since there are many points of regulation of protein expression following transcription. In these cases, the time point of the biopsy could be decisive and make the results congruent: an increase in SIRT1 protein was observed at 120 min of recovery, but at 30 min it remained unchanged [26], which coincides with the lack of change in the SIRT1 mRNA in skeletal muscle samples taken immediately after loading [29].

In addition to considering the influence of the methodology followed in each investigation on the results obtained, it is important to note that the results will vary depending on the type of participants. Radak et al. reported that the expression of SIRT1 mRNA increased in skeletal muscle samples from sedentary subjects taken immediately after a single exercise load compared to active subjects in whom there was no difference in mRNA [29].

In most studies, besides studying the effect of exercise on sirtuins, the impact on molecules or metabolic pathways related to sirtuins has also been analyzed. Guerra et al. [26] showed that sprint exercise alone elicits Thr172-AMPKα phosphorylation, which is compatible with the increase in SIRT1 protein previously reported by these authors, where it was shown that AMPK can increase SIRT1 [38]. Following this line, it is known that SIRT1 deacetylates and increases the transcriptional activity of PGC-1α [39], however, Guerra et al. did not find changes in PGC-1α protein. The researchers suggested that exercise-induced AMPK phosphorylation alone is not sufficient to increase PGC-α protein expression in the recovery period, but it could also be because PGC-1α protein was determined in the total muscle and not in the nucleus, where, as will be described below, other authors have reported changes.

A constant observation is that acute exercise does not increase either the protein or the mRNA of SIRT3 [26,27,28,29]. Nevertheless, it was observed that PGC-1α mRNA increased, without changes in SIRT3 mRNA, as an effect of acute exercise [27]. Analyzing these findings, it is possible to suggest that, more than by methodological differences or by missed changes of SIRT3, the change in PGC-1α was possibly due to an increase in SIRT1, which was not measured. Another possibility mentioned by Edget et al. (2016), is that a single exercise load is not a sufficient stimulus to increase the genetic expression of SIRT3, but repeated stimuli could cause an increase in SIRT3 genetic expression, and subsequently in the content of the protein [28]. However, there is no conclusive explanation for the observed decrease in SIRT3 mRNA [28,29].

Only Dumke et al. have analyzed the effect of an exercise load for three consecutive days. They found that the intervention caused acute increases in mRNA of the upstream elements as Pgc-1α and Sirt1, and chronic increases in the downstream targets as cytochrome C and CS, thus increasing mitochondrial biogenesis and oxidative enzyme capacity [30].

The observations derived from the effect of consecutive loads on the SIRT1 are close to the pattern observed in the training interventions with either HIIT or aerobic training. In general, research has found that several sessions of HIIT increase the protein or activity of SIRT1 and along with it the PGC-1α protein, either in the whole muscle or in the cell nucleus [31,32,33]. Nevertheless, some results seem contrary because it has been found that the SIRT1 protein does not change or even decreases, although its activity increases. In this case, the authors comment on the possibility of a transient increase in SIRT1 protein after exercise, although this increase is followed by a prolonged decrease that becomes evident with more than 2 weeks of training [32,33]. It has been described that the maximum activity of an enzyme can increase without increasing the protein content. The authors point out that the increase in SIRT1 activity despite the decrease in protein, strongly suggests that modifications to the SIRT1 protein occur during training and persist for prolonged periods [33].

The PGC-1α is a transcriptional co-activator that plays a crucial role in coordinating mitochondrial gene transcription. In most interventions with HIIT it was found that the nuclear PGC-1α protein increased, which indicates that PGC-1α is active [40]. Greater nuclear PGC-1α promotes or maintains an increase in mitochondrial biogenesis via increased co-activation of transcription factors linked to mitochondrial gene expression [31].

HIIT also increases the activity of COX, CS [29,30,31], β-hydroxyacyl-CoA dehydrogenase (β-HAD) [32,33] and the protein content of COXII, COXIV, TFAM and glucose transporter type 4 (GLUT4) [31,32]. Additionally, Edgett et al. also measured the effect of chronic sprint-interval training (SIT) on SIRT3, PGC-1α and leucine rich pentatricopeptide repeat containing (LRP130), which is a novel transcriptional coactivator proposed to promote the expression of mitochondrial encoded genes, resulting in increases in oxidative phosphorylation and fatty acid oxidation [41,42,43]. The researchers found no changes in protein content, however, individual changes in protein expression of LRP130, SIRT3, and PGC-1α were positively correlated at several time points—these results suggest that the regulation of these proteins may be coordinated in human skeletal muscle [27]. In all of the above, the HIIT-induced changes provide evidence that SIRT1 is mainly involved in regulating the increase of mitochondrial oxidative capacity and PGC-1α-mediated mitochondrial proliferation during human skeletal muscle training. In addition, it is interesting to note that although the activation or protein of AMPK was not determined in any of the interventions with HIIT described in Table 1, it has been reported that a session with four intervals of 30 s of high intensity separated by 4 min of rest increases phosphorylation of AMPKα1/α2 immediately after exercise and PGC-1 mRNA after 3 h recovery [44]. This indicates a relationship between HIIT-induced SIRT1/PGC-1α changes and AMPK activation.

Research on the effects of aerobic training has focused on analyzing the response of SIRT3. It is generally observed that SIRT3 protein increases in the muscle of sedentary subjects after several training sessions [34,36]. Brandauer et al. found no changes in the SIRT3 protein [35]. This could be due, on the one hand, to the fact that the participants were active, unlike the other investigations that analyzed sedentary subjects and, on the other hand, to the differences in the time point at which the biopsy was performed or the number of sessions performed. In our work, which analyzed the effect of aerobic training on SIRT3 in skeletal muscle of sedentary overweight adolescents, we found that the SIRT3 and PGC-1α proteins increased [36], which is in accordance with the axis AMPK-PGC-1α-SIRT3. It has been suggested that this axis improves mitochondrial function and gene expression to adapt to metabolic changes [45,46,47].

As already described, SIRT3 deacetylates key molecules that include elements of the TCA cycle and proteins involved in oxidative phosphorylation [16,48,49] as well as elements that manage reactive oxygen species (ROS) [47]. A potential mitochondrial signaling cascade response involving SIRT3 and forkhead box O3 (FOXO3A)-dependent transcription of catalase and MnSOD has been proposed [50]. MnSOD and catalase directs the conversion of superoxide to water and oxygen in sequential steps [51]. Besides, the activation of MnSOD is enhanced by SIRT3-dependent deacetylation at K122, allowing the cell to scavenge ROS [17]. Although Brandauer et al. did not find an increase in SIRT3 after endurance training, they observed that the MnSOD protein increased significantly [35]. These researchers complemented their research with an animal model (mice wild-type, with AMPKα2kinasa dead and with AICAR treatment) and concluded that AMPK and PGC-1α regulate protein abundance of SIRT3 and MnSOD in skeletal muscle in response to exercise training.

The findings of Johnson et al. also contributed to the ROS management line. Their participants were young and elderly sedentary people: the endurance training had an age-dependent response. The results showed that the young participants increased SIRT3 and MnSOD proteins without changes in the catalase, in addition, in the young individuals also increased the NAMPT protein, an upstream regulator of SIRT3. On the other hand, older adults after endurance training also increased the SIRT3 protein, although in a smaller proportion than the young, it was also observed that the enzymatic activity of MnSOD and catalase increased; NAMPT protein did not increase with training, but when analyzing the chronic effect of exercise (> 4 years), a higher content of this protein was observed in active than in sedentary older adults. The researchers concluded that endurance training increases muscle and mitochondrial antioxidant capacity in young and adult individuals [34]. The effects of aerobic training are similar to those already described in subjects who have exercised regularly throughout their lives [19,21].

We tested the effect of 12 weeks of resistance training in obese youths on the content of SIRT3 protein. The SIRT3 remained unchanged, suggesting that this protein might not be activated during resistance training because this type of training preferentially uses phosphocreatine and muscle glycogen for ATP synthesis in the sarcoplasm. This means that resistance training does not induce sufficient metabolic stress to activate oxidative phosphorylation, and consequently, the metabolic pathways that induce the activation of SIRT3 [37]. Other authors have reported that AMPKα2 activity increases immediately after an acute load of resistance exercise, however, after 2 h this effect decreases [52] and is possibly insufficient to activate the AMPK-PGC-1α-SIRT3 axis.

## 4. Conclusions and Perspectives

In conclusion, the effects of exercise on SIRT1 and SIRT3 in human skeletal muscle will depend on the type of exercise (Figure 1). The effect of acute exercise has been studied in both SIRT1 and SIRT3; the results indicate that SIRT1, and not SIRT3, is activated after a single load of exercise, apparently as an effect of increased phosphorylation of AMPK. The increase in SIRT1 leads to an increase in PGC-1α. The effect of exercise training on sirtuins has been studied with several sessions of HIIT or aerobic exercise. Investigations with HIIT have focused on analyzing the effect on SIRT1—these investigations provide evidence that the benefits of this type of training are regulated through the increase in SIRT1 and PGC-1α; however, there is little evidence on the effect on both sirtuins.

Meanwhile, the effect of interventions with aerobic training has been studied only on SIRT3. The results indicate that after several sessions of aerobic training there is an increase of SIRT3 in sedentary individuals and increases in PGC-1α. In addition, NAMPT also increases as an effect of exercise, allowing the synthesis of NAD+, the main substrate of sirtuins.

Due to the methodological complexity, the investigations have focused on some part of the metabolic pathway. The information we currently have is complemented by the results of all these investigations, even the animal and cellular models. It is known that exercise activates each one of the sirtuin pathways in different ways, but the outcome is common: to improve the mitochondrial health, which places them as therapeutic targets for the treatment of metabolic diseases such as obesity, insulin resistance and type 2 diabetes. So, in order to identify exercise training that can efficiently achieve greater improvement in these metabolic states, it will be necessary to perform more research that evaluates the effect of different types of exercise on both sirtuins and the health benefits in metabolically compromised individuals of different ages and different fitness level.

## Figures and Tables

**Figure 1 ijms-20-02717-f001:**
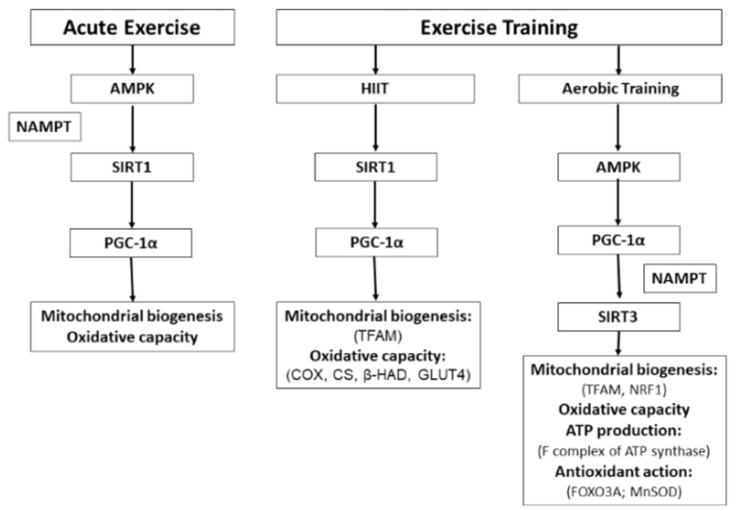
Main observed effects of different types of exercise on SIRT1 and SIRT3 in human skeletal muscle and the metabolic pathways involved. The molecules for which there is direct evidence derived from exercise interventions in humans appear in parentheses. AMPK, AMP-activated protein kinase; β-HAD, β-hydroxyacyl-CoA dehydrogenase; COX, cytochrome C oxidase; CS, citrate synthase; FOXO3A, forkhead box O3; GLUT4, glucose transporter 4; HIIT, high-intensity interval training; MnSOD, manganese superoxide dismutase; NAMPT, nicotinamide phosphoribosyltransferase; NRF1, nuclear respiratory factor 1; PGC-1α, peroxisome proliferator-activated receptor gamma coactivator-1α; TFAM, mitochondrial transcription factor A. The arrows indicate activation or increase.

**Table 1 ijms-20-02717-t001:** Effects of different types of exercise on SIRT1, SIRT3 and PGC-1α expression.

Type of Exercise	Mode	SIRT1	SIRT3	PGC-1α
Acute	Wingate test [26]	↑ (Prot)	-	= (Prot)
	SIT [27]	-	= (mRNA)	↑ (mRNA)
	Stationary bicycle [28]	-	↓(mRNA) = (Prot)	-
	Treadmill [29]	↑ (mRNA) (*S) = (mRNA)	= (mRNA) (*S) ↓(mRNA)	-
Successive Bouts	3 bouts of 3 h cycling [30]	↑ (mRNA)	-	↑ (mRNA)
HIIT	6 sessions low-volume [31]	↑ (Prot)	-	= Total (Prot) ↑ Nuclear (Prot)
	7 sessions [32]	↑ (act) = (Prot)	-	↑ Nuclear (Prot) ↑ Total (Prot)
	18 sessions [33]	↓(Prot) ↑ (act)	-	↑ (Prot)
	24 sessions SIT [27]	-	= (Prot)	= (Prot)
Aerobic Training	24 sessions Treadmill [34]	-	↑ (Prot) (*S)	-
	15 Sessions one-legged knee extensor [35]	-	= (Prot)	-
	36 sessions Cycle ergometer [36]	-	↑ (Prot) (*S) (OW)	↑ (Prot) (*S) (OW)
Resistance Training	36 sessions 11 exercises for the major muscle groups [37]	-	= (Prot) (*S) (OW)	= (Prot) (*S) (OW)

Results observed in the vastus lateralis muscle sample of trained or recreationally healthy young people unless it is specified that they were sedentary (* S) or overweight (OW). Act, activity; HIIT, high-intensity interval training; Prot, protein content; SIT, sprint-interval training.

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
