# Peer review of "Exercise and Sirtuins: A Way to Mitochondrial Health in Skeletal Muscle"

_ijms, 2019, doi:10.3390/ijms20112717_

Reviewer 1 Report

In the present study, the authors reviewed the effect of exercise on skeletal muscle mitochondrial health focusing on sirtuins. This review is important and well-documented, but there are some points should be considered.

 1. Include “skeletal muscle” in the Title for better understanding the contents of this review.

2. Please comment about that SIRT3 mRNA decreases after acute exercise (ref. 26, 27).

3. Regarding the effect of resistance training on SIRT3, the authors have commented that unchanged SIRT3 level by resistance exercise is attributed to insufficient metabolic stress (line 248-249). However, it has been reported that resistance exercise activates AMPK (see below ref). In general, AMPK activation means that metabolic stress is occurred. Please revise including this point. In addition, HIIT is also sufficient to activate AMPK (see below ref). Pease comment the association between HIIT-induced SIRT1/PGC1a changes and AMPK.

 Dreyer HC, Fujita S, Cadenas JG, Chinkes DL, Volpi E, Rasmussen BB. Resistance exercise increases AMPK activity and reduces 4E-BP1 phosphorylation and protein synthesis in human skeletal muscle. J Physiol. 2006 Oct 15;576(Pt 2):613-24. doi: 10.1113/jphysiol.2006.113175. Epub 2006 Jul 27. PubMed PMID: 16873412; PubMed Central PMCID: PMC1890364.

Gibala MJ, McGee SL, Garnham AP, Howlett KF, Snow RJ, Hargreaves M. Brief intense interval exercise activates AMPK and p38 MAPK signaling and increases the expression of PGC-1alpha in human skeletal muscle. J Appl Physiol (1985). 2009 Mar;106(3):929-34. doi: 10.1152/japplphysiol.90880.2008. Epub 2008 Dec 26. PMID: 19112161

Author Response

Point 1: Include “skeletal muscle” in the Title for better understanding the contents of this review.

Response 1: We have changed the title included "skeletal muscle" 

 Point 2: Please comment about that SIRT3 mRNA decreases after acute exercise (ref. 26, 27).

Response 2: We have added information (lines 183 – 187)

 Point 3 : Regarding the effect of resistance training on SIRT3, the authors have commented that unchanged SIRT3 level by resistance exercise is attributed to insufficient metabolic stress (line 248-249). However, it has been reported that resistance exercise activates AMPK (see below ref). In general, AMPK activation means that metabolic stress is occurred. Please revise including this point. In addition, HIIT is also sufficient to activate AMPK (see below ref). Pease comment the association between HIIT-induced SIRT1/PGC1a changes and AMPK.

Response 3 : We have added information with the recommended references Dreyer et al (lines 264-266) and Gibala et al (lines 220-224).

Reviewer 2 Report

Sirtuins are NAD+-dependent histone deacetylases that have emerged as key regulators of many functions including metabolism, cell growth and apoptosis, as well as control of the aging process. Recent studies have demonstrated that some types of exercise affect the expression and activity of sirtuins in several tissues. This review focuses on the effects of exercise on  on SIRT1 and SIRT3 of human skeletal muscle. The authors also discussed how sirtuin mediated regulation affects PGC1-a and AMPK as mentioned in previous report and mitochondrial biogenesis. The review is well written but missing many wonderful works.

So before going for acceptance of this manuscript, I would suggest to authors to revisit literature search related to physical exercise and sirtuins and further include in this manuscript. For e.g. Villanova et.al. talking first time about influence of age and physical exercise on sirtuin activity in humans, Bayod et.al. talking long-term physical exercise and their effects on Sirt1 pathway. One very important publication, Rahman et al. 2014, suggesting the mammalian sirt3 is an important deacetylase for regulation of ATP synthase b for ATP production.

Author Response

Point 1: So before going for acceptance of this manuscript, I would suggest to authors to revisit literature search related to physical exercise and sirtuins and further include in this manuscript. For e.g. Villanova et.al. talking first time about influence of age and physical exercise on sirtuin activity in humans, Bayod et.al. talking long-term physical exercise and their effects on Sirt1 pathway. One very important publication, Rahman et al. 2014, suggesting the mammalian sirt3 is an important deacetylase for regulation of ATP synthase b for ATP production.

 Response 1: About this topic we have included Lanza (2008) and Villanova (2013) (lines 123, 130-132). Also, we included your recommended reference Beyod et al (76-78).

Finally we decided don´t included Rahman et al because they show results in  Drosophila melanogaster, however, we included Vassilopoulos et al because they reported results in mice and humans.